# Antibiotic Resistance and Phylogeny of *Pseudomonas* spp. Isolated over Three Decades from Chicken Meat in the Norwegian Food Chain

**DOI:** 10.3390/microorganisms9020207

**Published:** 2021-01-20

**Authors:** Even Heir, Birgitte Moen, Anette Wold Åsli, Marianne Sunde, Solveig Langsrud

**Affiliations:** 1Nofima, Norwegian Institute of Food, Fisheries and Aquaculture Research, P. O. Box 210, N-1431 Ås, Norway; birgitte.moen@nofima.no (B.M.); anette.asli@nofima.no (A.W.Å.); solveig.langsrud@nofima.no (S.L.); 2Department of Animal Health and Food Safety, Section of Food Safety and Emerging Health Threats, Norwegian Veterinary Institute, P.O. Box 750 Sentrum, N-0106 Oslo, Norway; marianne.sunde@vetinst.no

**Keywords:** antibiotic resistance, *Pseudomonas*, phylogeny, poultry, whole-genome sequencing

## Abstract

*Pseudomonas* is ubiquitous in nature and a predominant genus in many foods and food processing environments, where it primarily represents major food spoilage organisms. The food chain has also been reported to be a potential reservoir of antibiotic-resistant *Pseudomonas*. The purpose of the current study was to determine the occurrence of antibiotic resistance in psychrotrophic *Pseudomonas* spp. collected over a time span of 26 years from retail chicken in Norway and characterize their genetic diversity, phylogenetic distribution and resistance genes through whole-genome sequence analyses. Among the 325 confirmed *Pseudomonas* spp. isolates by 16S rRNA gene sequencing, antibiotic susceptibility profiles of 175 isolates to 12 antibiotics were determined. A subset of 31 isolates being resistant to ≥3 antibiotics were whole-genome sequenced. The isolates were dominated by species of the *P. fluorescens* lineage. Isolates susceptible to all antibiotics or resistant to ≥3 antibiotics comprised 20.6% and 24.1%, respectively. The most common resistance was to aztreonam (72.6%), colistin (30.2%), imipenem (25.6%) and meropenem (12.6%). Resistance properties appeared relatively stable over the 26-year study period but with taxa-specific differences. Whole-genome sequencing showed high genome variability, where isolates resistant to ≥3 antibiotics belonged to seven species. A single metallo-betalactmase gene (*cphA*) was detected, though intrinsic resistance determinants dominated, including resistance–nodulation (RND), ATP-binding cassette (ABC) and small multidrug resistance (Smr) efflux pumps. This study provides further knowledge on the distribution of psychrotrophic *Pseudomonas* spp. in chicken meat and their antibiotic resistance properties. Further monitoring should be encouraged to determine food as a source of antibiotic resistance and maintain the overall favorable situation with regard to antibiotic resistance in the Norwegian food chain.

## 1. Introduction

*Pseudomonas* has been described as one of the most ubiquitous bacterial genera in the world, and it has been found in a wide variety of outer environments (e.g., seawater, fresh water, soil, rhizosphere, bird droppings, cyanobacterial mat samples in Antarctica, Atacama desert), human environments (e.g., drinking water, kitchens, oil fields, thermal power plants, sludge, food production plants), clinical specimens (e.g., pus, lungs with cystic fibrosis, urinary tract infection, septicemia) and infected plants (e.g., leaf spots, brown blotch disease in mushrooms; [1]). The genus is as diverse as its habitats, with nearly 200 species and a complex phylogeny [2]. Foodborne psychrotrophic *Pseudomonas* spp. have high metabolic versatility, adaptive capacity and growth abilities at low temperatures, supporting their dominating prevalence in various parts of the food chain [3,4]. This genus includes the opportunistic human pathogen *P. aeruginosa*, which is a troublesome clinical, antibiotic-resistant bacterium, though with variable and generally low prevalence in foods and food environments [5]. For the chicken meat industry, *Pseudomonas* is primarily an important spoilage organism. The plucking and scalding processes usually reduce the number of bacteria, including *Pseudomonas,* to low numbers. However, as shown in multiple studies across several decades and continents, remaining *Pseudomonas* from the raw materials or from processing surfaces, chilling air or water can grow rapidly in the food product, especially under aerobic storage [6]. At the end of the shelf life, *Pseudomonas* dominates and spoils the meat through proteolytic, lipolytic, saccharolytic and biosurfactant processes [7].

A range of different methods with variable accuracy and discriminatory power have been and are at present used to identify *Pseudomonas*, making it difficult to draw general conclusions about which species are most prevalent on chicken meat. In a study from 2017, using both 16S rRNA gene sequencing and Illumina MiSeq to describe the microbiota of chicken meat, 21 different species were found, with a dominance of *P. weihenstephanensis* (new species 2016) and *P. congelans* [8]. Most likely, a variety of *Pseudomonas* may be present on the raw materials, but different growth abilities between *Pseudomonas* lead to selection of some groups over others. In early studies from Norway and Belgium, using carbohydrate and fatty acid profiles, it was reported that most isolates from spoiled poultry were similar to *P. fluorescens*, *P. lundensis* and *P. fragi* [9]. The same species were dominant on spoiled, marinated, poultry from Chile, when using specific PCR for identification to species level [7]. In a US study, *P. fluorescens*, *P. putida*, *P. chloroaphis* and *P. syringae* were isolated from poultry carcasses after storage, using fatty acid profiling for identification [6]. The increasing use of genomic analyses will likely provide more accurate species identification and thereby improved understanding of *Pseudomonas* phylogeny and characteristics in the food chain [2,10,11,12].

The predominance and persistence of *Pseudomonas* spp. in foods and on food processing surfaces have also been related to the ability of these microorganisms to form a biofilm, which enhances their tolerance to adverse conditions including antimicrobial treatments [12,13]. Studies have increasingly reported the role of various parts of the food chain as possible reservoirs for antibiotic-resistant bacteria and resistance genes. This includes reports on antibiotic-resistant *Pseudomonas* spp. in chicken and other meats [14,15], milk and dairy products [11,12,16] and vegetables [17]. Furthermore, the extraordinary ability of *P. aeruginosa* to acquire new mechanisms of resistance through horizontal acquisition of resistance genes carried on plasmid, integrons or transposons has been demonstrated [18]. Antibiotic-resistant *Pseudomonas* spp. could therefore act as potential reservoirs of antibiotic resistance in the food chain, with the ability of resistance gene transfer to other bacteria including potential pathogens.

In Norway, antimicrobial resistance has since 2000 been monitored through the Norwegian monitoring program for antimicrobial resistance in feed, food and animals, NORM-VET (http://www.vetinst.no/overvaking/antibiotikaresistens-norm-vet). The prevalence of antibiotic-resistant bacteria in the animal food production chain in Norway is low compared to most other European countries [19]. Use of antimicrobial agents in the Norwegian livestock production is also low, as demonstrated by the ESCVAC (European Medicines Agency, European Surveillance of Veterinary Antimicrobial Consumption) report [20]. However, the application of selective methods for the detection of antibiotic-resistant bacteria showed that 50% of investigated turkey meat samples harbored quinolone-resistant *E. coli*, although at very low quantitative levels [21,22]. Selective methods also identified high prevalence (around 40%) of resistant *E. coli* in fecal samples from broilers and retail chicken meat, respectively [23], prevalence that was reduced according to similar data from 2016 [24]. Transferrable plasmid-mediated resistance has been reported for both these resistance traits [21,25]. In NORM-VET, susceptibility testing of *E. coli* and *Enterococcus* spp. is generally used as an indicator of the occurrence of antimicrobial resistance in the bacterial population. The recent history of common though unexpected detection of antibiotic-resistant *E. coli* in the Norwegian poultry production chain, along with the known high prevalence of *Pseudomonas* spp. in the food chain and their reported potential for resistance gene transfer, support the need for more knowledge on the role of *Pseudomonas* spp. in antibiotic resistance prevalence, distribution and resistance gene reservoirs in the food chain. The aim of the current study was to determine the occurrence of antibiotic resistance in psychrotrophic *Pseudomonas* spp. in retail chicken in Norway and characterize their genetic diversity, phylogenetic distribution and resistance genes through whole-genome sequence analyses. *Pseudomonas* strains isolated over a time span of 26 years (1991–2017) from the retail poultry chain in Norway were included.

## 2. Materials and Methods

### 2.1. Sample and Strain Collection

*Pseudomonas* isolates included in this study (n = 325) were from separate collections from raw poultry in the period 1991 (one collection) and 2014–2017 (three collections; Table 1). Collection 1 and 2 were already established Nofima in-house strain collections, whereas collection 3 and 4 were established as a part of this work to obtain an assortment of *Pseudomonas* spanning a long time period and different products and sources. All the poultry products were obtained from the southeastern region of the Viken county in Eastern Norway.

Collection 1 from 1991 was from six flocks of chicken carcasses slaughtered at two Norwegian slaughterhouses in a period from February to July 1991 [9]. *Pseudomonas* were isolated one day after slaughter and after 9 days of refrigerated storage. Isolates identified as *Pseudomonas* spp. from colony morphology on Plate Count Agar (Difco) containing 1% NaCl, Gram staining, motility, catalase activity, fatty acid profiling and carbon source assimilation tests were stored frozen in 20% glycerol stocks [9]. To obtain an overall representative collection and reduce the probability of selecting clonal isolates, only *Pseudomonas* spp. isolates with different carbon assimilation profiles from the same flock and sampling date were chosen and 16S rRNA gene sequenced (n = 97).

Collection 2 (2014) included isolates (n = 34) collected from eight raw chicken products from six manufacturers. The products were bought in local supermarkets and kept cold until analysis. Samples were obtained by surface swabbing (FLOQSwabs, COPAN Diagnostics Inc., Ca, USA) of a 2 × 10 cm^2^ area of the chicken surface (legs, fillets with and without skin), vortexing the swab in 3 mL peptone water and plating on Pseudomonas CFC agar (Pseudomonas agar base with Pseudomonas CFC selective agar supplement (Oxoid, ThermoFisher Scientific, Oslo, Norway) and incubation at 20 °C, 48 h. Up to five presumptive *Pseudomonas* spp. were randomly collected from each of the eight products.

Collection 3A and 3B (2016) originated from the Norwegian surveillance program on antibiotic resistance in poultry products (NORM/NORM-VET 2016; Usage of Antimicrobial Agents and Occurrence of Antimicrobial Resistance in Norway). From a total of 60 samples of fresh retail chicken products in the period September to December 2016, *Pseudomonas* spp. isolates were collected by parallel plating of stomached chicken samples (25 g) (on CHROMagar Pseudomonas, Paris, France) for 20 °C, 48 h, without (Collection 3A) and with 0.5 mg/L ciprofloxacin (collection 3B). The latter was used for selective isolation of *Pseudomonas* spp. resistant to ciprofloxacin. From each sample, four colonies representing, if possible, different morphologies of presumptive *Pseudomonas* were chosen. A total of 113 and 36 confirmed *Pseudomonas* spp. isolates were collected from agar plates without and with CIP, respectively.

Collection 4 (2017) included samples from one of the slaughterhouses where the isolates from 1991 (Collection 1) were collected. The slaughterhouse was visited three times and surface swab samples from six chicken carcasses per visit were vortexed and plated on CHROMagar Pseudomonas (20 °C, 48 h). A total of 60 presumptive *Pseudomonas* spp. isolates were collected by picking 2–4 colonies with *Pseudomonas* spp.-like color and morphology per carcass sample. All pure culture isolates were stored in 15% glycerol at −80 °C.

### 2.2. 16S rRNA Gene Analyses

A total of 443 isolates were partial 16S rRNA gene sequenced to exclude non-*Pseudomonas* genus isolates from the collection. Partial amplification and sequencing of the 16S rRNA gene were performed using universal primers (Nadkarni et al. 2002). DNA was isolated by boiling single bacterial colonies resuspended in 50 µL Tris-EDTA buffer at 99 °C for 10 min. Bacterial debris were removed by centrifugation, and 30 µL of the supernatant containing the DNA was transferred to a new tube, of which 1µL was used as template in the polymerase chain reaction (PCR). Amplification was performed using 0.2 µmol/L (each) primer, 12.5 µL of Platinum hot start PCR 2X Mastermix (Invitrogen, ThermoFischer Scientific) in a total volume of 25 µL on a Veriti 96-well Thermal Cycler (Applied Biosystems (ThermoFisher)). The cycling conditions were: 94 °C 2 min, then 30 cycles of denaturing at 94 °C for 30 s, annealing at 60 °C for 30 s, extension at 72 °C for 30 s, and a final extension at 72 °C for 7 min. The PCR products were purified before sequencing, using ExoSap-IT (Applied Biosystems (ThermoFisher)). Sequencing PCR was carried out using the Big Dye^®^ Terminator v1.1 Cycle Sequencing Kit (Applied Biosystems) according to the manufacturer’s instructions with the forward universal 16S rDNA primer. The sequencing reactions were carried out in 25 cycles of 96 °C for 15 s and 60 °C for 4 min. A BigDye XTerminator Purification Kit (Applied Biosystems, Foster City, CA, USA) was used according to the manufacturer’s recommendations to clean up the sequencing reactions and sequencing was performed on an ABI PRISM 3500 Genetic Analyzer (Applied Biosystems).

### 2.3. 16 S rDNA Phylogenetic Analyses

The CLC Main Workbench (versions 20.0.4 and 7.8.1) was used to inspect and trim sequences (quality trimming limit 0.02). The trimmed sequences were aligned and cut to equal length (288 bp) and used to construct a phylogenetic tree (neighbor joining tree with Jukes–Cantor distance measure and bootstrap (100 replicates)). The taxonomy was identified using BLAST 16S rRNA (bacterial and archaea type strains) database (including 21,264 sequences, accessed September 12th 2020).

### 2.4. Antimicrobial Susceptibility Testing

Isolates for susceptibility testing (n = 175) were selected according to the 16S rRNA gene sequence phylogenetic grouping of the isolates. When several isolates within one flock or sample had the same 16S rRNA gene sequence, only one isolate was tested to avoid the bias that would occur if including clonal strains (Table 1). The selection criteria provided a common basis for selecting isolates from the different sample collections for MIC analyses and subsequent comparative analyses and statistics.

Susceptibility to antibiotics was determined by a broth microdilution method using Sensititre Gram Negative DKMGN Plates for non-fastidious Gram-negative isolates (ThermoFisher Scientific, Oslo, Norway). Inoculum preparations were according to manufacturer’s recommendations. In short, fresh colonies on Tryptic Soya Agar (Oxoid) were suspended in Sensititre Demineralized water (ThermoFisher Scientific) to 0.5 McFarland using a nephelometer. Then, 10 ml of this suspension was mixed in 11 ml Sensititre Cation Adjusted Mueller–Hinton Broth w/TES buffer (ThermoFisher Scientific) and inoculated (50 ul) into each well of the plate using a multi-channel pipette. The plate was incubated at 30 °C for 18–24 h. The plates were read using the Sensititre Manual Viewbox. The minimum inhibitory concentration (MIC) was determined, defined as the lowest concentration (mg/L) of antibiotic of which no growth was observed. *P. aeruginosa* ATCC 27,853 was used as a quality control strain. Breakpoints used were in accordance with EUCAST *Pseudomonas* spp. breakpoints when available, else *P. aeruginosa* breakpoints. Five of the 17 antibiotics in the panel were not considered relevant due to reported natural, intrinsic resistance for *Pseudomonas* spp. (EUCAST Clinical Breakpoint Tables v. 10.0) and were disregarded. MIC tests were repeated for isolates showing resistance to three or more relevant antibiotics.

### 2.5. Integron-PCR

Integrons were detected using PCR. Primers for detection of the cassette regions of class 1 and class 2 integrons were hep58 and hep59 (White et al. 2000), and hep74 and hep51 (White et al. 2001), respectively. PCR amplifications were performed in 25-ul reactions using 0.2 uM of each primer and Invitrogen Platinum Hot Start PCR Master Mix (2X) (ThermoFischer Scientific) on a Veriti 96-well Thermal Cycler (Applied Biosystems (ThermoFisher Scientific)). Bacterial DNA was isolated by QIAprep Spin Miniprep Kit (Qiagen, Oslo, Norway) in accordance with the supplier’s recommendations and 1 µl was used as template. PCR amplicons were detected by agarose gel electrophoresis (0.7% agarose with GelRed Nucleic Acid Gel strain (Biotium, CA, USA)). Positive and negative controls were used in all PCR reactions. *E. coli* Se 131 (Acc. No. AJ238350) and *E. coli* 620-11 [26] were used as positive controls for detection of the cassette regions of class 1 and class 2 integrons, respectively.

### 2.6. Whole-Genome Sequencing (WGS)

All isolates resistant to three or more antibiotics and the three class 1 integron-positive isolates were subjected to whole-genome sequencing (WGS). The strains were grown on tryptone soy agar (TSA) plates (Oxoid) at 30 °C overnight. Colonies from the plates were transferred into 5 mL tryptone broth and incubated in a rotatory incubator at 150 rpm and 30 °C, overnight. One ml of the overnight culture was centrifuged at 13,000× *g* for 5 min, and the pellet was used for DNA purification. The cells were lysed for 40 s at 6 m/s using lysing matrix B tubes containing 0.1 mm silica spheres and a FastPrep instrument (both MP Biomedicals), and DNA was isolated using the DNeasy blood and tissue kit (Qiagen) in accordance with the supplier’s recommendations. The DNA concentration was measured using Quant-iT PicoGreen ds DNA Assay Kit (Invitrogen, ThermoFisher Scientific) and Synergy H1, microplate reader (BioTek, VT, USA).

Libraries for genome sequencing were prepared using the Nextera XT DNA Sample Preparation Kit (Illumina, San Diego, CA, USA) and sequenced using paired-end (PE) 2 × 300 bp reads on a MiSeq instrument (Illumina). The isolates were sequenced in three separate MiSeq runs (Run 1: MCS v2.6.2.1, MSR v2.6.2, RTA v1.18.54; Run 2 and 3: MCS v3.1.0, LRM v2.0.1, RTA v1.18.54; Appendix A).

#### Analyses of WGS Data

Prior to de novo assembly, removal of adapter sequences and q15 quality trimming was performed using fastq-mcf (Version: 1.05) from the ea-utils package [27]. Genome assembly was performed using SPAdes v 3.13.0 [28] with the careful option and six k-mer sizes (21,33,55,77,99,127). Contigs with size <500 bp and with coverage <5 were removed from the assemblies. The genome assemblies were evaluated using QUAST v 5.0.2 [29] and the sequences were annotated using PROKKA 1.14.6 [30]. Average coverage of the filtered assemblies was calculated using BBMap v36.x (Bushnell B.—sourceforge.net/projects/bbmap/).

Core genome phylogenetic analysis was performed in Roary [31] using the Prokka annotation; a 95% nucleotide identity cut-off was applied for inclusion in the core genome. Sequences were aligned using MAFTT [30,32] and neighbor joining phylogeny was reconstructed in CLC Main Workbench 20.0.4, with Juker–Cantor approximation and bootstrap (100 replicates). The resulting phylogenetic tree was combined with the gene_presence_absence table generated by Roary to construct a core genome phylogenetic tree with gene presence/absence matrix (using roary_plots.py).

ABRicate (version 1.0.1; https://github.com/tseemann/abricate) was used to search for antibiotic resistance genes (database: NCBI AMRFinderPlus; CARD; Resfinder; ARG-ANNOT and MEGARES 2.00) and plasmids (database PlasmidFinder). All databases used were updated in April 2020. We also included *Pseudomonas* spp. antimicrobial resistance genes reported by Meng et al. [11] and Quintiery et al. [12] and annotations obtained using Prokka in the search for resistance genes. PubMLST (Public databases for molecular typing and microbial genome diversity) and Ribosomal Multilocus Sequence Typing (rMLST) were used for species identification (https://pubmlst.org/species-id; [33]). rMLST is an approach that indexes variation in the 53 genes encoding the bacterial ribosome protein subunits (*rps* genes) as a means of integrating microbial taxonomy and typing. The rMLST database was accessed November 16th 2020. Phylogenetic relationships between our isolates and other *Pseudomonas* isolates were established by employing the NDtree (nucleotide difference tree; [34,35,36]). In total, 28 assemblies were selected from available representative *Pseudomonas* genomes (n = 223) as well as the reference genome *P. aeruginosa* PAO1 and downloaded from the NCBI microbial genome database https://www.ncbi.nlm.nih.gov/genome/ (per December 8th 2020; Appendix A) and submitted together with our 31 isolates (contigs) to NDtree service (version 1.2) at https://cge.cbs.dtu.dk/services/NDtree/. The representative genomes were selected based on potential relatedness to our isolates. *P. putida* and *P. aeruginosa* were chosen as out-groups. The tree was visualized in CLC Main Workbench.

The draft genome assemblies for the 31 *Pseudomonas* isolates have been deposited in the public domain (NCBI BioProject PRJNA685056). Minor adjustments (additional adapter trimming) were performed in some genomes before being released by the NCBI. The raw data have been deposited in the NCBI Short Read Archive (SRA; BioProject PRJNA685056).

### 2.7. Statistical Analyses

Isolates included in the comparative analyses and statistics for MIC analyses were according to the selection criteria specified above, if not otherwise stated. Statistical analyses of differences in resistance prevalence between isolates of different sample collections were conducted by Fisher’s exact test with significance level of *p* < 0.05.

## 3. Results

### 3.1. Overall Identification and 16S rRNA Gene Phylogenetic Analyses of Bacteria from Chicken Samples

Presumptive *Pseudomonas* isolates (n = 443) from four chicken meat sample collections, spanning a time period of 26 years, were collected. Good-quality 16S rRNA gene sequences (partial, 288 bp) were obtained from 360 isolates, of which 325 were confirmed to belong to the *Pseudomonas* genus. Other genera identified were *Acinetobacter*, *Aeromonas*, *Enterobacter*, *Janthinobacterium*, *Morganella*, *Shewanella*, *Stenotrophomonas* and *Vibrio*. These were discarded from further analyses.

A phylogenetic overview of the 325 *Pseudomonas* isolates is shown in Figure 1. The 16S sequencing grouped the isolates into a limited number of subclades. Interestingly, isolates within each of the four sample collections were in general distributed across the whole phylogenetic tree.

Of the 325 isolates, 322 belonged to the *P. fluorescens* lineage, of which, respectively, 284, 16 and 13 isolates grouped into the *P. fluorescens* group (including *P. koreensis* subgroup), *P. syringae* group and *P. chlororaphis* group [10]. There were nine isolates in this lineage that could not be assigned to a group (*P. silesiensis, P. turukhanskensis* and *P. versuta*). Three isolates grouped in the *P. aeruginosa* lineage; one was matched in BLAST with *P. caeni* and two with *P. flavescens* (the three isolates in the bottom of the phylogenetic tree, Figure 1). The prevalence of the different species within the different collections is shown in Appendix A. The carbon profiling of isolates from collection 1 corresponded well to the 16S rRNA gene clusters, with a few individual exceptions—however, with different species assignment (Appendix A). For example, all isolates with carbon profile A assigned to *P. fluorescens* biovar I-1 [9] were in the same 16S cluster (*P. lactis*). Likewise, all isolates with carbon profile C (assigned to *P. lundensis*) were assigned to the same 16S cluster (*P. weihenstephanensis*).

### 3.2. Phenotypic Antimicrobial Resistance and Presence of Integrons

Susceptibility to 12 antibiotics was tested for a selection of 175 isolates (Figure 1, Table 2). For nine antibiotics, MIC values above the defined resistance breakpoints for *Pseudomonas* spp. were found. The most common resistance was to aztreonam (72.6%), colistin (30.2%), imipenem (25.6%) and meropenem (12.6%). None of the isolates were resistant to aminoglycosides (amikacin, gentamicin, tobramycin) while only 2.3% (four isolates) were resistant to the fluoroquinolone ciprofloxacin. Resistance to the cephalosporin ceftazidime (10.2%) was found for a single isolate only (0.6%), when ceftazidime was used in combination with the β-lactamase inhibitor avibactam, indicating that beta-lactamases inhibited by avibactam have a role in ceftazidime resistance.

Collection 1 (Year: 1991; n = 53) contained the highest proportion of isolates susceptible to all antibiotics (30.2%; n = 16) while collection 4 (Year: 2017; n=38) contained the lowest and highest proportion of susceptible (5.3%; n = 2) and multi-resistant (28.9%; n = 11) isolates (Figure 2), respectively. For isolates collected after selective plating on CIP-containing agar plates (collection 3B), the proportion of isolates susceptible to all antibiotics or resistant to ≥3 antibiotics was 0% and 53.8%, respectively. The highest number of resistance traits was obtained for a single isolate from collection 3B, being resistant to seven antibiotics.

Excluding *Pseudomonas* spp. collected from CIP plates, 22.2% (n = 36) of the isolates were susceptible to all tested antibiotics while resistance to ≥3 antibiotics was found in 21.6% (n = 35). Although the results indicated that less *Pseudomonas* collected in the period 2014–17 were susceptible to antibiotics (18.3%) than those collected in 1991 (30.2%), the differences were not statistically significant (*p* = 0.08). Similarly, although the proportion of isolates resistant to three or more antibiotics was higher in 2014–2017 collections than the 1991 collection (23.9% vs. 17.0%), the difference was not statistically significant (*p* = 0.42). However, a higher proportion of aztreonam resistance was evident in isolates collected in 2014–2017 (78.9%) compared to isolates from 1991 (54.7%; *p* = 0.003). For colistin, the opposite was observed, with 23.9% and 39.6% resistant isolates in these two groups (*p* = 0.04).

There appeared to be taxa-specific differences in antibiotic resistance, with certain taxa having a high proportion of susceptible and resistant isolates, respectively (Figure 1, Table 3).

Three isolates (collection 3A (*P. corrugata*) and 3B (*P. gessardii* and *P. weihenstephanensis*)) harbored genes for *intI*1 class 1 integrons although with empty cassettes. No *intI*2 class 2 integrons were detected.

### 3.3. WGS of a Subset of Resistant Pseudomonas spp.

A total of 31 isolates were resistant to several (three or more) antibiotics and/or contained integrons and these were whole-genome sequenced. The average genome size of the isolates was 6,307,185 bp (largest genome: 6,811,860 and smallest genome: 5,804,763). The probable copy numbers of rRNA operons determined by Prokka were between 4 and 8 (Appendix A). One isolate (MF7453) had a repeat region identified as a CRISPR-cas1 region.

The rMLST analyses using WGS data allowed a more precise taxonomic identification compared to 16S rRNA gene sequencing, separating the isolates into seven different species: *P. fluorescens* (>98% match, n = 7), *P. brenneri* (>96% match, n = 5), *P. haemolytica* (n = 4), *P. carnis* (n = 2), *P. lactis* (n = 1), *P. psychrophilia* (n = 1), *P. simiae* (n = 1). The remaining 10 isolates could not be assigned to a specific species. To get a more comprehensive view of the taxonomy of our isolates, we compared the isolates with 29 genomes selected from the NCBI microbial genomes using NDtree (representative sequences for comparable species in addition to *P. putida* and *P. aeruginosa* as out-groups were chosen). The rMLST taxonomic identification and the NDtree clustering showed good correlations (Appendix A) and revealed similar clusters as for the 16S rRNA gene tree—however, with another taxonomic identification. For example, isolates identified to *P. canadensis* based on partial 16S rRNA gene were identified to *P. haemolytica* by rMLST and NDtree, and isolates identified to *P. lactis* by 16S rRNA gene were separated into three species based on WGS: *P. fluorescens*, *P. carnis* and *P. lactis*. Moreover, two isolates classified as *P. fluorescens* by rMLST clustered together with *P. carnis* and were reassigned to *P. carnis* by NCBI after submission (based on average nucleotide identity (ANI)).

Further WGS data analyses revealed a high genomic variability, with estimated 42,492 genes in the pan-genome, 291 core (genes present in >99% of strains), 300 soft-core, 10,307 shell and 31,594 cloud genes. A phylogenetic tree was generated from the core gene alignment and combined with the gene_prescense_absence table from Roary (Figure 3).

Resistance determinants reported to be plasmid-bound were seldom identified. Plasmid detection and analysis was not the scope of this study and further investigations (e.g., with long read sequencing) are required to rule out the presence of plasmids in the isolate collections.

### 3.4. Antimicrobial Resistance Determinants in Pseudomonas spp.

Reported resistance determinants of ten different antibiotic classes, cationic disinfectants as well as a range of multidrug resistance genes were found among the 31 isolates subjected to genome sequencing. In total, more than 70 genes associated with antibiotic resistance were identified (Table 4).

Seven genes often associated with β-lactam resistance were detected, including the instrinsic *ampC* gene, present in all 31 genome sequenced isolates. Other genes, *mrcA* and *pbpC*, encoding penicillin-binding proteins were present in 31 and 10 isolates, respectively. A single isolate, MF6787 (Collection 3A), identified to *P. brenneri* according to rMLST, contained the metallo-beta-lactamase gene *cphA*, but did not show higher resistance to meropenem and imipenem than other isolates in the same cluster/taxa or in many distantly related isolates. Resistance determinants providing specific resistance to the monobactam aztreonam were not determined, although it was the antibiotic for which the highest proportion of isolates was resistant. Other genes associated with β-lactam resistance encoded the MexAB-OprM efflux system and the outer membrane protein OprD present in 29 and 31 of the strains, respectively. A gene homolog (*pbpC*) to the target gene of aztreonam in *P. aeruginosa*, *ftsI*, encoding a penicillin-binding protein (Jorth et al. 2017), was identified in 10 of the WGS isolates. The *pbpC* encodes a penicillin-binding protein but their role in aztreonam resistance in *Pseudomonas* spp. is unknown.

Six genes were detected (*emrA*, *lpxA*, *lpxD*, *pgsA*, *phoP*, *phoQ*) that were reported to be associated with resistance to colistin, a last-resort lipopeptide antibiotic for the treatment of MDR Gram-negative bacterial infections. The genes *emrA* and *phoP* encoding a colistin resistance protein and a transcription regulator, respectively, were most prevalent according to Prokka and were detected in 28 and 31 of the genome sequenced strains.

A number of genes encoding multidrug efflux systems were detected. The majority of these belonged to the resistance–nodulation–cell division (RND) family of efflux pumps that is associated with membrane fusion proteins and outer membrane porins (e.g., TolC, OprM) involved in tolerance to a wide spectrum of antimicrobials and toxic compounds. Among the most prevalent RND multidrug transporter genes detected were *acrB*, *mexB*, *ttgB*. Genes encoding efflux proteins of other families included *norM, mepA,* (multi-antimicrobial extrusion protein family; MATE), *emrB*, *mdtD, qacA* (the major facilitator superfamily; MFS) and *qacC, emrE* (small multidrug resistance family; SMR). The *qac* genes, only found in one strain each, encode proteins involved in reduced susceptibility to quaternary ammonium compound disinfectants.

Resistance genes towards aminoglycosides were almost absent. The *neo* gene conferring resistance to neomycin was the single aminoglycoside resistance gene detected in 18 isolates.

## 4. Discussion

Several studies have reported *Pseudomonas* spp. to be a dominant bacteria on chicken carcasses in abattoirs [37,38,39] and also to be prevalent and a main cause of spoilage in a range of foods, e.g., red meats, milk and fresh produce [12,17,40,41]. Dominant species reported on chicken meat include *P. fluorescens*, *P. fragi* and *P. putida*, but limited data are available on the genome diversity, taxonomy as well as phenotypic and genotypic antibiotic resistance of *Pseudomonas* spp. in the food chain. This may partly be due to the extensive heterogeneity and complex phylogeny of *Pseudomonas* and the fact that discriminatory methods for species assignment are not always used [10]. A challenge is also the ability of non-*Pseudomonas* bacteria to grow on *Pseudomonas* selective agar media such as Pseudomonas CFC agar and CHROMagar Pseudomonas, as illustrated in the present study, where 35 of 360 isolates collected from these media belonged to other bacterial genera.

Whole-genome sequencing of the antibiotic-resistant isolates of the current study illustrated the overall large genetic heterogeneity of the *Pseudomonas* isolates, with more than one megabase difference between the largest and smallest genome and where the core genome represented only a minor fraction of the extensive pan-genome (Figure 3). Comparison of these isolates with selected genomes from the NCBI microbial genomes using NDtree further illustrated the high genomic diversity within *Pseudomonas*, and the phylogenetic tree would probably look different if we included more isolates for each species (Appendix A). It has been assumed that the increased use of WGS or multilocus sequence analysis (MLSA) will result in a substantial increase in the number of *Pseudomonas* species and also split the current *Pseudomonas* genus into several genera or subgenera [10]. This also shows the challenge in comparing, e.g., prevalence and phylogenetic distribution of strains and species between studies that have used different methods with various degrees of discriminatory power.

Resistance to one or more antibiotics and known resistance determinants occurred in *Pseudomonas* from chicken meat. Nevertheless, resistance to specific antibiotics varied greatly. Within the group of beta-lactam antibiotics, the highest proportions of resistance were to the monobactam aztreonam (72.6%) followed by the carbapenems imipenem (25.6%) and meropenem (12.6%). Colistin-resistant isolates reached 30.2% while all isolates were susceptible to aminoglycosides except neomycin.

There appeared to be taxa-specific differences (based on partial 16S rRNA gene) in resistance properties, with isolates of certain taxa being more susceptible (e.g., *P. deceptionensis*, *P. fragi* and *P. weihenstephanensis*) while other taxa dominated among isolates resistant to ≥3 antibiotics (e.g., *P. canadensis* and *P. lactis*). Among the three most dominating species, *P. weistephaniensis*, *P. lactis* and *P. gessardi*, distinct differences were found, where none of the two latter species were totally susceptible, whereas 79% of the former were susceptible to all antibiotics. The antibiotic resistance profiling indicated no dramatic changes in the overall antibiotic resistance of raw chicken-associated *Pseudomonas* spp. in the 26-year study period. However, differences in the proportion of isolates being resistant to aztreonam and colistin between the two time periods (1991 vs. 2014–2017) were indicated. Since it was not only the year of sampling that differed between the collections, but also the storage after slaughter and methodology for isolation of strains, it is difficult to conclude the specific reasons for differences found. In the samples from 1991, there was a high proportion of *P. weistephaniensis*, with a generally low frequency of resistance, and this could partly explain why less resistance was observed. Variations in resistance between *Pseudomonas* species have also been reported by others [16]. The distribution of resistance according to phylogeny also reflects that the majority of isolates subjected to WGS were within a limited number of phylogenetic clades. One should be aware that partially different methods for sampling and cultivation of the isolates were applied that may have influenced the somewhat variable distribution of species apparent in the different collections (Appendix A). The observed differences and the limited data available should promote further studies on surveillance, resistance levels and mechanisms of antibiotic resistance among *Pseudomonas* spp. and other resident bacteria in the food chain.

As expected, a higher proportion of isolates collected by selective plating on CIP-containing agar were resistant to antibiotics than isolates collected without CIP. Surprisingly, only three out of the 13 CIP-selected isolates showed resistance to CIP in the MIC test and seven isolates were inhibited by the CIP levels in the selective agar plates they were isolated from. The observed resistance was difficult to explain by the presence of specific resistance genes. No isolates harbored plasmid-mediated quinolone resistance genes (PMQR) as detected in *Pseudomonas* spp. from milk, water and seafood in other countries [11,42,43]. More likely, chromosomal mutations (i.e., in the genes *gyrA*, *gyrB*, *parC*, *parE*) induced by low-level CIP exposure and altered expression of multidrug transporters can explain how these isolates were able to grow on agar with CIP in the initial selection. It has been reported that exposure to ciprofloxacin has been associated with the development of multidrug resistance [44]. Fluoroquinolones are mutagenic and also among the most common substrates of multidrug efflux pumps [43].

Resistance prevalence to the monobactam aztreonam was high, but similar to reported resistance in *Pseudomonas* spp. from milk (60.5%; [11]); healthy animals (50.3%, [45]) and bulk tank milk (32%, [16]). There is limited knowledge on the prevalence and distribution of beta-lactamase genes in members of the *Pseudomonas* genus other than *P. aeruginosa*. A total of 19 mutated genes were associated with aztreonam resistance in a recent study on *P. aeruginosa* [46]. Chromosomal mutations conferring increased AmpC expression and efflux through the MexAB-OprM efflux system were the most important mechanisms. Genes encoding these proteins were also present in nearly all genome sequenced isolates in this study, but mutation analyses were not performed. No specific monobactam resistance genes have been reported in non-*aeruginosa Pseudomonas* species. In *P. aeruginosa*, a penicillin-binding protein FtsI [47] is the primary target for aztreonam and mutations in the *ftsI* gene may provide aztreonam resistance. A homolog to FtsI, encoded by *pbpC*, was identified in 10 of the genome-sequenced isolates in the current study, but their role in aztreonam resistance is unknown and can, in any case, not explain the overall aztreonam resistance. Further studies are needed to determine mechanisms of resistance, reported potential fitness costs [46] and factors selecting aztreonam resistance in *Pseudomonas* sp. non-*aeruginosa* in the food chain.

Resistance to the carbapenems imipenem (25.6%) and meropenem (12.6%) was significant although limited both in the prevalence and resistance levels compared to resistance to these antibiotics reported in *Pseudomonas* spp. from milk in China [11]. Lower proportions of carbapenem-resistant strains from food were reported by others but few strains were studied [17,48]. The study of Meng et al. [11] reported 14 different β-lactamase genes while the current study detected seven genes associated with β-lactam resistance. The only metallo-beta-lactamase gene detected in the present study was *cphA*, encoding a subclass B2 MBL and present in a single *P. brenneri* (rMLST) isolate (MF6787). The CphA protein exerts resistance to the carbapenems meropenem and imipenem but has no activity towards monobactams such as aztreonam. The amino acid (aa) sequence had a 100% match with a subclass B2 metallo-β-lactamase from *P. brenneri* (GenBank OAE14554.1). It also shared high similarity with the PFM-1 (90%; MN065826), PFM-2 (96%; MN080496) and PFM-3 (96%; MN080497), identified in *P. fluorescens* complex isolates also from chicken. Other isolates within the same cluster/taxa also showed high tolerance/resistance to meropenem and imipenem although they did not harbor *chpA*. Therefore, additional mechanisms are most likely involved and the role of the *chpA* for tolerance in *P. brenneri* is uncertain.

Only a few carbapenem resistance genes have been reported in *P.* sp. non-*aeruginosa* but the number of studies is limited [15,49]. High resistance levels (up to 256 mg/L) have been explained by the acquisition of carbapenemase genes [15], possibly through transformation, as *P. fluorescens* strains are spontaneously transformable at high frequency [50]. Plasmid-encoded carbapenemase genes have also been reported in *Pseudomonas* sp. non-*aeruginosa* species (Quientiery et al., 2019). No such genes were detected in the present study. Further studies are needed to determine the role of *Pseudomonas* spp. as a reservoir and vector for carbapenem resistance gene transfer to infectious *P. aeruginosa* [51].

β-lactam resistance in *Pseudomonas* spp. may also occur from mutations in the intrinsic cephalosporinase AmpC (Juan, Torrens, Oliver Review 2017) and the penicillin-binding protein Pbp1a encoded by *mrcA*. Both *ampC* and *mrcA* were present in all 31 genome-sequenced isolates. Mutations in *mrcA* have been reported to have a role in increased *ampC*-mediated β-lactam resistance in *Stenotrophomonas maltophilia* [52]. Their role in β-lactam resistance needs to be further elucidated in *Pseudomonas* spp.

Altered expression/function in RND efflux systems (e.g., TtgABC, MexAB-OprM, AcrAB-TolC; [15,53]) and porin channels such as OprD [11,17,54,55] have also been reported. Genes encoding these proteins were all apparent although in different distributions in our genome-characterized strains. Finding clear correlations between phenotypic resistance and gene mutations/expression levels of, e.g., OprD can be difficult due to high genetic diversity in homologous genes within and between *Pseudomonas* species [15,56].

The frequent occurrence of resistance to carbapenems found in the present and other studies is worrying because they are considered as last-resort antibiotics for the treatment of multidrug-resistant human pathogens. Carbapenems have never been licensed for food-producing animals in any country [57]; nevertheless, carbapenem resistance in foodborne bacteria is a growing concern. Data combining whole genome analyses, phenotypic resistance and relevant metadata may be an important further step to unravel these complex mechanisms of resistance and factors affecting resistance development.

Interestingly, we detected colistin resistance in almost one third of the isolates and with a higher proportion of resistance among isolates collected almost 30 years ago (in 1991; Collection 1) compared to the isolates collected in the period 2014–2017. The emerging transferable plasmid-encoded colistin resistance determinants (*mrc*) were, however, not detected. Colistin resistance is largely unknown in non-aeruginosa *Pseudomonas* although both acquired and intrinsic resistance have been reported, including a naturally resistant *Pseudomonas* species (*P. mallei*; [58]). Additional resistance mechanisms include activation of the regulatory gene operon *phoPQ* and further activation of *arnA*, leading to lipid A modifications associated with colistin resistance [59]. The *arnA* gene was present in 30 isolates while the *phoPQ* operon was present in only nine of the 19 colistin-resistant WGS isolates according to annotations by Prokka. Overexpression of multidrug efflux systems (e.g., AcrAB-TolC, NorM, EmrAB) and downregulation of porin OprD have also been reported as colistin resistance mechanisms (see review by Moubareck [58] and references therein).

It has been demonstrated that exposure of *P. aeruginosa* to increasing levels of QAC-based disinfectants contributed to higher tolerance to polymoxin B, structurally similar to colistin (polymoxin E; [60]). Exposure to another biocide frequently applied in food processing environments, sodium hypochlorite, also resulted in a statistically significant increase in resistance to various antibiotics including both colistin and meropenem [61]. Multidrug efflux mechanisms can be induced by membrane damaging agents [62]. However, correlations between increased expression of efflux pumps and their contribution to resistance remain unclear [63]. The phenomenon of heteroresistance with subpopulations able to multiply in the presence of colistin has also been reported [64]. However, we observed no “skipped wells” phenomenon (wells with no growth, although growth still occurs at higher concentrations) in our susceptibility tests. Further studies are needed to reveal the diverse and complicated mechanisms of colistin resistance in *Pseudomonas*.

Several genes encoding efflux transporters involved in reduced susceptibility to QACs were detected, although with low prevalence. This included the *emrE* (two isolates), and one isolate each containing *qacA*, *qacC* and *mdfA*. The encoded protein of the latter has been extensively characterized in *E. coli* and confers resistance to a broad variety of unrelated cytotoxic compounds including monovalent and divalent cations, CIP and also neutral chloramphenicol. The *mdfA*-containing isolate (MF6776) was the second most resistant isolate in our study collection, being resistant to five antibiotics. However, this and other recent studies such as Meng et al. [11] have shown that it is not possible to fully deduce phenotypic resistance properties to the presence of resistance genes.

The presence of integrons is of great concern as these elements can capture and spread antibiotic resistance gene cassettes. Studies have reported over 30% prevalence of class 1 integrons in *Pseudomonas* spp. obtained from communal and hospital (treated or untreated) aquatic effluents [65]. In the present study, only three isolates (1.7%) carried a class 1 integron *intI*1 gene, all with empty gene cassettes. The integrons were associated with a weak promoter and an integron-associated recombination site (*attI*1; [66]). Empty integrons have also been found in Gram-negative bacteria from farm animals and fish farming environments [67,68]. The results indicate a low prevalence of integrons, but under environmental pressures, these structures could integrate antibiotic resistance gene cassettes [69].

The relatively stable resistance levels observed over nearly three decades indicate a low antibiotic selective pressure in the Norwegian chicken meat chain. This is supported by the very low level of antibiotic usage in the Norwegian broiler production in recent years, where only few flocks are treated yearly [70,71]. Antibiotic consumption data in poultry production from earlier years (e.g., 1990s) are more scarce but Grave et al. reported total usage of phenoxymethylpenicillin (the most commonly used antibiotic in Norwegian poultry production) to be 8 kg in 1991, according to sales data [72]. Phenoxymethylpenicillins and amoxicillin have in recent years been the most commonly applied antibiotics, although at low levels throughout the study period [71]. Of interest, the observed prevalence of resistance to aztreonam, imipenem, meropenem and colistin is not likely to be associated with therapeutic use of these antibiotics in poultry production in Norway. The former three have never been used on poultry while available data (from 1993) on colistin show no use of this in animals [73].

## 5. Conclusions

There is still limited knowledge on the prevalence and mechanisms of antibiotic resistance in the foodborne microbiota. *Pseudomonas* is the dominant bacterial genus in many food processing environments and they are important spoilage organisms in foods stored aerobically and cold. The overall increased consumption along with the history of antibiotic resistance associated with chicken meat implies the relevance of investigating *Pseudomonas* spp. and their antibiotic resistance properties in chicken meat products.

The vast majority of *Pseudomonas* in chicken meat belonged to psychrotolerant species of the *P. fluorescens* group. These are rarely associated with human infections, but *P. fluorescens* can cause bloodstream infections [74] and are parts of the human microbiota.

Although a significant proportion of the isolates from chicken were resistant to certain antibiotics (meropenem, imipenem, colistin, aztreonam), a very limited number of genes encoding resistance to specific antibiotics were detected and transferable resistance mechanisms were not evident. Only one specific carbapenem resistance gene was detected in a single isolate. Resistance properties varied according to phylogeny and are likely dominated by intrinsic mechanisms (e.g., multidrug efflux). Therefore, from the present study, it seems that *Pseudomonas* spp. are not significant carriers of resistance determinants that could be disseminated to pathogens [70]. However, with the increased need to understand the role of the food chain in antimicrobial resistance, monitoring the presence of resistance genes and their host organisms in various food chain compartments to outline strategies to minimize antibiotic resistance dissemination is essential, also in environments with low antibiotic selection. Understanding potential non-antibiotic selection pressures affecting the development and dissemination of antimicrobial resistance in the food chain also merits further studies.

## Figures and Tables

**Figure 1 microorganisms-09-00207-f001:**
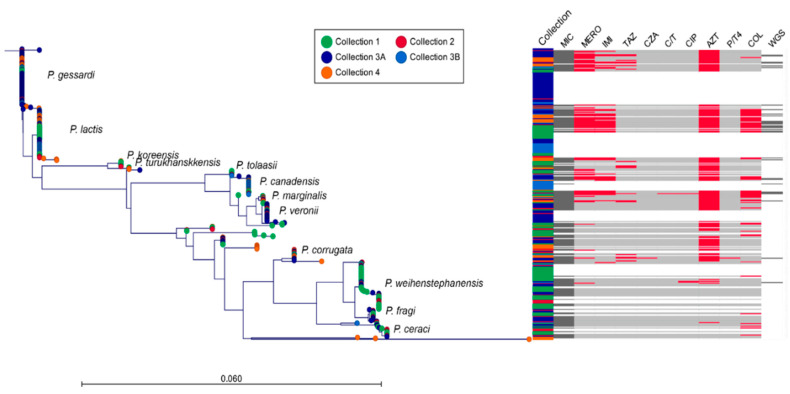
Phylogenetic distribution of *Pseudomonas* spp. isolates (n = 325) based on partial 16S rRNA gene (288 bp) sequencing. Assignment of the isolates to presumptive species and to the strain collections is indicated. The subsets of isolates subjected to MIC (n = 175) and WGS (n = 31) analyses are shown. Isolates with resistance to the specific antibiotics are indicated in red. Antibiotic abbreviations: MERO (meropenem), IMI (imipenem), TAZ (ceftazidime), CZA (ceftazidime/avibactam), C/T (ceftolozane/tazobactam), CIP (ciprofloxacin), AZT (aztreonam), P/T4 (piperacillin/tazobactam), COL (colistin). Antibiotics for which all isolates were sensitive (amikacin, gentamicin, tobramycin; according to Table 2) were not included.

**Figure 2 microorganisms-09-00207-f002:**
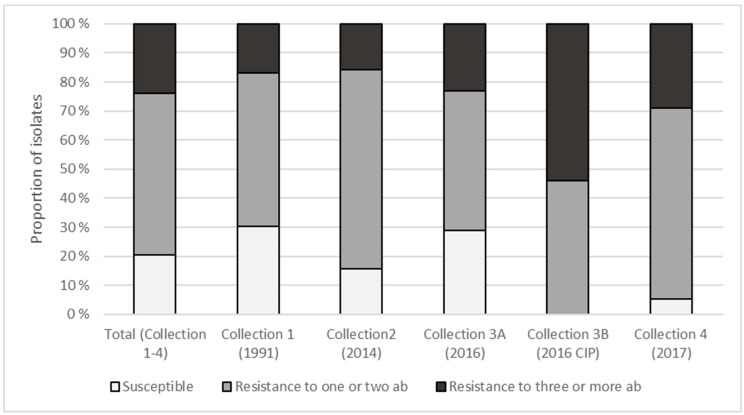
Proportion of antibiotic-susceptible and resistant isolates (total numbers n = 175) for all collections. See Table 2 for the 12 antibiotics tested.

**Figure 3 microorganisms-09-00207-f003:**
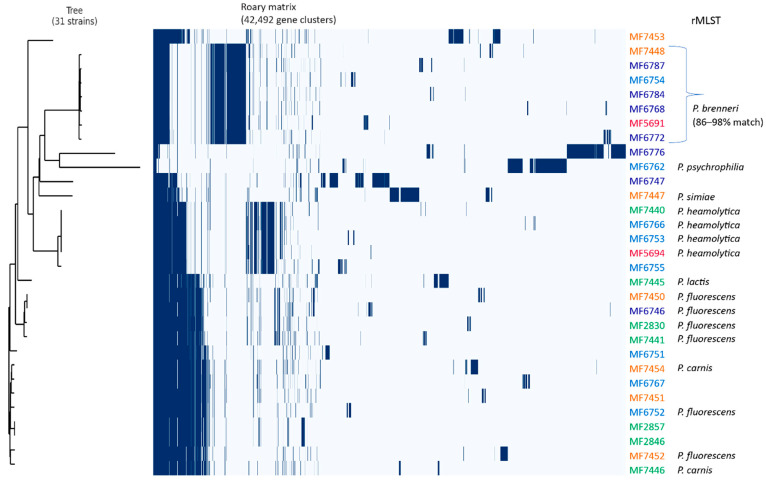
Core gene alignment tree. The presence or absence of 42,492 aggregated pan-genome genes in 31 isolates is shown. Blue denotes that a gene is present in the corresponding isolate (identified by isolate name to the right). The isolate names are colored according to the collection number (see Figure 1). Species assignment is shown according to rMLST for isolates with 100% match (except for *P. brenneri*).

**Table 1 microorganisms-09-00207-t001:** Overview of chicken product samples and number of analyzed *Pseudomonas* spp.

Sample Collection (Reference)	Sample Type	Year	Chicken Samples	16S rRNA Gene	MIC ^a^	WGS ^b^
1 [9]	Cold stored Carcasses	1991	20	97	53	7
2 (This study)	Raw fillet at retail	2014	8	34	19	2
3A (This study)	Raw fillet at retail	2016	60	113	52	7
3B (This study)	Raw fillet at retail	2016	53 ^c^	36	13	8
4 (This study)	Raw carcass in slaughterhouse	2017	18	45	38	7
Total				325	175	31

^a^ Numbers are MIC (minimal inhibitory concentration)-tested isolates selected and determined in accordance with selection criteria detailed in Material and Methods. ^b^ Breakpoint value for imipenem was changed from ≥ 8 mg/L to ≥ 4 µg/ml during the study period (EUCAST). For determination of isolates to WGS, the former breakpoint value of imipenem (≥ 8 mg/L) was applied in the analyses. The number of isolates selected for WGS (whole genome sequencing) was therefore fewer than the number of isolates resistant to three or more antibiotics reported elsewhere in this article. ^c^ 53 of the 60 chicken samples of sample collection 3A (plated on CHROMagar Pseudomonas without CIP; ciprofloxacin) were also plated on CHROMagar Pseudomonas with 0.5 mg/L CIP and were thus defined Collection 3B.

**Table 2 microorganisms-09-00207-t002:** Minimum inhibitory concentrations (MIC) and antimicrobial resistance in *Pseudomonas* spp. (n = 175) isolated from raw chicken in Norway.

Antibiotic	Resistance (%)	Distribution (%) of MIC Values (mg/L) ^a^										
		0.06	0.12	0.25	0.5	1	2	4	8	16	32	64
Amikacin	0.0							99.4	0.6			
Gentamicin	I.E. ^b^				96.6	1.7	1.1	0.6				
Tobramycin	0.0					1.7	98.3					
Meropenem	12.6		22.3	6.9	10.9	8.0	10.9	16.6	12.0	8.6	4.0	
Imipenem	25.6				19.4	20.6	21.1	13.1	11.4	13.1	1.1	
Ceftazidime	10.2				1.1	5.7	29.7	33.7	19.4	5.1	5.1	
Ceftazidime/Avibactam	0.6				1.1	8.6	46.9	39.4	3.4	0.6		
Ceftolozane/Tazobactam	0.6				20.6	52.0	21.7	5.1	0.6			
Ciprofloxacin	2.2	54.9	30.3	9.7	2.9	1.1	1.1					
Aztreonam	72.6					0.6	0.0	0.6	13.7	12.6	12.0	60.6
Piperacillin/Tazobactam	0.6					1.1	12.0	42.9	34.9	8.6	0.6	
Colistin	30.2			1.1	23.4	36.6	8.6	7.4	5.1	17.7		

^a^ Vertical, bold lines denote breakpoint values for resistance for each antibiotic (EUCAST Clinical Breakpoint Table v. 10.0). White fields denote concentration range of each tested antimicrobial agent. Susceptibility profiles of the 175 isolates to 12 antibiotics are presented in Appendix A. MIC values higher than the highest concentration tested are given as the lowest MIC value above the range. MIC values equal to or lower than the lowest concentration tested are given as the lowest concentration tested. ^b^ I.E. = insufficient evidence that *Pseudomonas* spp. are good targets for therapy with the agent (Ref. EUCAST)—no breakpoint value defined.

**Table 3 microorganisms-09-00207-t003:** Antibiotic resistance of 175 *Pseudomonas* spp. The percentage of isolates within each species with MIC above the breakpoint value for 0, 1, 2, 3, 4, 5 or 7 antibiotics (ab = 0 to ab = 7) is shown.

Best Match (Species) ^1^	Isolates (N)	MIC (N)	Proportion of Isolates (%)						
			ab = 0	ab = 1	ab = 2	ab = 3	ab = 4	ab = 5	ab = 7
*P. brassicacearum*	1	1	0.0	0.0	100	0.0	0.0	0.0	0.0
*P. brenneri*	1	1	0.0	100	0.0	0.0	0.0	0.0	0.0
*P. caeni* ^2^	1	0							
*P. canadensis*	22	11	0.0	0.0	9.1	54.5	27.3	0.0	9.1
*P. cerasi*	16	10	50.0	50.0	0.0	0.0	0.0	0.0	0.0
*P. corrugata*	13	9	0.0	66.7	22.2	0.0	0.0	11.1	0.0
*P. deceptionensis*	2	2	100	0.0	0.0	0.0	0.0	0.0	0.0
*P. flavescens*	2	2	50.0	50.0	0.0	0.0	0.0	0.0	0.0
*P. fragi*	13	8	87.5	12.5	0.0	0.0	0.0	0.0	0.0
*P. gessardii*	68	26	0.0	53.8	15.4	11.5	19.2	0.0	0.0
*P. helleri*	6	4	25.0	75.0	0.0	0.0	0.0	0.0	0.0
*P. helmanticensis*	4	3	0.0	100	0.0	0.0	0.0	0.0	0.0
*P. koreensis*	7	5	0.0	60.0	40.0	0.0	0.0	0.0	0.0
*P. lactis*	57	26	0.0	0.0	15.4	61.5	23.1	0.0	0.0
*P. marginalis*	11	11	0.0	18.2	72.7	0.0	9.1	0.0	0.0
*P. migulae*	11	10	0.0	100.0	0.0	0.0	0.0	0.0	0.0
*P. rhodesiae*	1	1	0.0	100	0.0	0.0	0.0	0.0	0.0
*P. silesiensis*	5	4	25.0	75.0	0.0	0.0	0.0	0.0	0.0
*P. tolaasii*	6	4	0.0	75.0	25.0	0.0	0.0	0.0	0.0
*P. turukhanskensis*	3	3	0.0	33.3	66.7	0.0	0.0	0.0	0.0
*P. veronii*	22	9	0.0	88.9	11.1	0.0	0.0	0.0	0.0
*P. versuta*	1	1	0.0	100	0.0	0.0	0.0	0.0	0.0
*P. weihenstephanensis*	52	24	79.2	20.8	0.0	0.0	0.0	0.0	0.0
SUM	325	175	20.6	40.6	14.9	14.3	8.6	0.6	0.6

^1^ Best match based on 288 bp og the 16S rRNA gene and the BLAST 16S rRNA (bacterial and archaea type strains) database. ^2^ This isolate was not viable when selected for MIC testing.

**Table 4 microorganisms-09-00207-t004:** Resistance determinants in *Pseudomonas* spp. (n = 31).

Antibiotic Class/Type of Antimicrobial	Phenotypic Resistance ^a^	Resistance Determinants (No. of Isolates) ^b^
*Antibiotic class*		
Aminocoumarine	Aminocoumarine	*mdtA* (31), *mdtB* (31), *mdtC* (30)
Aminoglycoside	NEO	*neo* (18)
β-lactam	IMI, MERO, β-lactam antibiotics	*ampC* (31), *ampD* (31), *cphA* (1), *pbpC* (10), *mrcA* (31), *mrcB* (9), *oprD* (31)
Bicyclomycin	BC	*bcr* (29)
Fluoroquionolones	CIP	*Bmr3* (2), *gyrA* (31), *gyrB* (31), *lfrA* (6), *mfd* (31), *parC* (31), *parE* (31)
Fosfomycin	FOS	*abaF* (31), *fosA* (13), *mdtD* (30), *mdtG* (13)
Fosmidomycin	FOSM	*fsr* (22)
Lipopeptides	COL	*arnA* (30)*, emrA* (28), *emrB* (27), *lpxA* (2), *lpxD* (4), *pgsA* (2), *phoP* (31), *phoQ* (9)
Macrolides	Macrolides, peptide antibiotics	*macA* (31), *macB* (31)
Tetracycline		*tetC* (7)
*Disinfectants and cationic agents*	QAC, EBR, various cationic agents	*emrE* (2), *qacA* (1), *qacC* (1)
*Multiple drugs* ^c^	Cationic peptides, fluoroquinolones, aminoglycosides	*acrA* (12), *acrB* (15), *acrE* (2), *acrF* (29) *aphA* (28), *aphB* (3)
	Carbenicillin, chloramphenicol, erythromycin, novobiocin, streptomycin and tetracycline	*arpC* (26)
	Various multidrugs	*bepC* (29), *bepE* (15), *bmr3* (2), *emrA* (31), *emrB* (30), *emrD* (5), *emrE* (2), *emrY* (24), *mdfA* (1), *mdtE* (16), *mdtK* (31), *mdtL* (28), *mdtN* (1), *mepA* (31), *mexA* (29), *mexB* (29), *mtrA* (19), *norM* (31), *oprJ* (5), *oprM* (29), *oqxB7* (31), *oqxB10* (27), *oqxB13* (1), *stp* (30), *tolC* (30)
	Aromatic hydrocarbons, fluoroquinolones, beta-lactams except imipenem	*ttgA* (31), *ttgB* (31), *ttgC* (31), *ttgI* (31)

^a^ Reported main phenotypic resistance traits are shown. Abbreviations: BC (Bicyclomycin), CIP (Ciprofloxacine), COL (colistin), EBR (ethidium bromide), FOS (fosofomycin), FOSM (fosmidomycin), IMI (imipenem), MERO (meropenem), NEO (neomycin), QAC (quaternary ammonium compounds). ^b^ Resistance determinants according to Prokka annotations. ^c^ Multidrug resistance genes associated with antibiotic and/or disinfectant resistance were included.

## Data Availability

Data are contained within the article or Appendix A.

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
