# Peer review of "Antibiotic Resistance and Phylogeny of Pseudomonas spp. Isolated over Three Decades from Chicken Meat in the Norwegian Food Chain"

_microorganisms, 2021, doi:10.3390/microorganisms9020207_

Round 1

Reviewer 1 Report

Interesting and well written paper.

I don’t have major issue except that the generalisability of the results of this study should be discussed and that there were no data on antibiotic use was gathered. Whether the latter has influence on the results? Also the link on human data should be discussed in the Discussion section.

Minor points:

Line 127. What part of chicken? Legs?

Norway is a big country. Please mention briefly how the geographic origins of these collections

Results. It is quite surprising that  (325/360) 90% was confirmed belonging to Pseudomonas genus. This should be commented in the Discussion section (‘wrong’ identification technique in the past? Or contaminated storage of the isoaltes? This should be also commented.

Figure 2 ‘proportion susceptible to all antibiotics’. Please also mention to which antibiotics tested.

Tabel 4. Could the resistant determinants be subdivided to the results of the rMLST analyses (or in other words, seven different species).

Author Response

Response to Reviewer 1 comments

Reviewer's general comments

Interesting and well written paper.

I don’t have major issue except that the generalisability of the results of this study should be discussed and that there were no data on antibiotic use was gathered. Whether the latter has influence on the results? Also the link on human data should be discussed in the Discussion section.

Authors response: We appreciate the positive response of the Reviewer and the suggestions to improve the manuscript.

We agree with the Reviewers comment on data on antibiotic use and have included input and references on this in the Introduction and in the last part of the Discussion (before Conclusion section). As the current manuscript is on antibiotic resistance of Pseudomonas in the chicken food chain, our input in the revised manuscript focuses on providing relevant data and references of antibiotic use in food production animals and in chicken in the study period (1991-2017; lines 659-671). The potential link of our findings to human data were also requested. This was indicated in the original manuscript and has been briefly extended in the revised manuscript (lines 593-594). We regard that further informative discussions on this without being too speculative would require human clinical data which was out of the scope of this study. Additional discussion to link the present data from the chicken chain to human data has therefore not been included.    

Reviewer Minor points and Authors response:

Line 127. What part of chicken? Legs?

Author response: The raw chicken products included both legs and fillets with and without skin. Information has been included in the revised manuscript. 

Norway is a big country. Please mention briefly how the geographic origins of these collections

Author response: Information on the geographic origin of the poultry has been included in paragraph. 2.1

Results. It is quite surprising that  (325/360) 90% was confirmed belonging to Pseudomonas genus. This should be commented in the Discussion section (‘wrong’ identification technique in the past? Or contaminated storage of the isoaltes? This should be also commented.

Author response: Partial 16S rRNA gene sequencing was used as a screening to verify and identify presumptive Pseudomonas isolates from the different collections. All the previously characterized isolates from 1991 were verified to be Pseudomonas. The other tree collections were obtained by picking and culturing  presumptive Pseudomonas isolates from agar media. Although the media used (CFC and CHROMagar Pseudomonas) are manufactured as selective for Pseudomonas, other bacteria have the ability to grow on these media. This was reflected in the analyses were almost 10 % (35 of 360) of the picked isolates were of other genera than Pseudomonas. This observation has been commented in the first paragraph of the Discussion section.  

Figure 2 ‘proportion susceptible to all antibiotics’. Please also mention to which antibiotics tested.

Author response: Information on antibiotics tested has been included in the Figure 2 legend.

Table 4. Could the resistant determinants be subdivided to the results of the rMLST analyses (or in other words, seven different species).

Author response: Among the 31 whole genome sequenced isolates (and thereby rMLST analysed isolates), 21 isolates were assigned to a total of seven Pseudomonas species according to rMLST analyses while for the remaining 10 isolates no clear species identifications were obtained. Connecting the data on resistance determinants to the identified species (rMLST analyses) as proposed by the reviewer will therefore be incomplete and inconclusive at this point. We agree that this type of information could be of interest when comparing a larger set of isolates than included in the current study. We therefore suggest to keep Table 4 in the original state and not to include another Table of limited informative value. Data on gene content of the individual strains can be accessed through the available accession numbers of the sequenced strains.  

Reviewer 2 Report

General comments

The work is well presented, and results are easy to follow. However, there are some concerns that need to be addressed. Also, it would be good to have an indication of the trend in antibiotic use in food animals in Norway over the decades. This could maybe explain the relatively stable resistance observed over the study period. If the use has decreased, it would be interesting to know why the resistance did not decrease and vise versa. If the use is constant, could the resistance arise from the use of other food additives that may contain antibiotic resistance-inducing substances like heavy metals?

Specific comments

Line 12-13: Pseudomonas is…..

Line 14: antibiotic-resistant (please correct throughout the manuscript)

Line 17 and 104: Delete “reservoirs”,

Line 25: Delete “in resistance properties”

Line 54: dominates and spoils

Line 62: “Pseudomonads” should not be in italics as it is not a scientific name

Line 88:  to most other European countries

Lines 166-167: Please be specific with the DNA isolation. It is not clear what “by lysis” means. Provide a reference if this was according to a reported method, to allow reproducibility.

Lines 168-172: Please specify the instrument used for the PCR. Also, did the authors include controls for their PCR?

Line 208-209: I suggest maintaining “antibiotics” rather than “antimicrobials” as antimicrobials could include other agents other than antibiotics.

Lines 215-223: Please provide the instrument used for the PCR and indicate the gel percentage and staining agent.

Line 232: …the pellet was used for DNA isolation using ………(mention the kit) following the manufacturer’s instructions.

In all the DNA extraction protocols, it would be good for the authors to mention how the purity and quantity of the extracted DNA was ascertain before performing the sequencing.

Line 506: The authors mention samples of 2001, which does not appear in the materials and methods (See Table 1).

Author Response

Response to Reviewer 2 comments

Reviewers General comments

The work is well presented, and results are easy to follow. However, there are some concerns that need to be addressed. Also, it would be good to have an indication of the trend in antibiotic use in food animals in Norway over the decades. This could maybe explain the relatively stable resistance observed over the study period. If the use has decreased, it would be interesting to know why the resistance did not decrease and vise versa. If the use is constant, could the resistance arise from the use of other food additives that may contain antibiotic resistance-inducing substances like heavy metals?

Author response: We thank the Reviewer for a thorough review and for a positive response. We also appreciate the suggestions to improve the manuscript. 

We agree with the Reviewers suggestion to include data on the usage of antibiotics in food production animals in Norway. This is important information and we have included relevant input and references  in the Introduction and Discussion (before Conclusions) sections. Detailed data on antibiotic usage is not easily available, especially not from decades ago.  Correlating antibiotic usage and resistance could therefore be speculative. However, we have included a sentence in the last part of the Discussion (lines 696-698) reflecting the importance of understanding non-antibiotic selection pressures in the development and dissemination of antibiotic resistance in the food chain.    

Reviewers Specific comments and author response

Line 12-13: Pseudomonas is…..

Author response: Corrected as suggested.

Line 14: antibiotic-resistant (please correct throughout the manuscript)

Author response: The term has been corrected throughout the manuscript

Line 17 and 104: Delete “reservoirs”,

Author response: Changes have been done according to reviewer suggestion

25: Delete “in resistance properties”

Author response: Done according to reviewer suggestion

Line 54: dominates and spoils

Author response: Corrected as suggested

Line 62: “Pseudomonads” should not be in italics as it is not a scientific name

Author response: "Pseudomonads" has been corrected to "Pseudomonas" and the italics has been kept.

Line 88:  to most other European countries

Author response: Corrected as suggested by the reviewer

Lines 166-167: Please be specific with the DNA isolation. It is not clear what “by lysis” means. Provide a reference if this was according to a reported method, to allow reproducibility.

Author response: DNA was isolated by a boiling method as specified in the revised manuscript.

Lines 168-172: Please specify the instrument used for the PCR. Also, did the authors include controls for their PCR?

Author response: The thermal cycler used has been included as requested. For 16S rRNA gene amplifications, positive controls were not included. PCR using specific primers and DNA from lysed bacteria will give a PCR product of known size. Positive controls were therefore not regarded to be essential in 16S rRNA gene amplification. Negative controls were used to identify possible contaminations during experimental set-up. 

Line 208-209: I suggest maintaining “antibiotics” rather than “antimicrobials” as antimicrobials could include other agents other than antibiotics.

Author response: We agree. Corrected as suggested

Lines 215-223: Please provide the instrument used for the PCR and indicate the gel percentage and staining agent.

Author response: We have included information on instrument for PCR, gel percentage and staining agent as requested.

Line 232: …the pellet was used for DNA isolation using ………(mention the kit) following the manufacturer’s instructions.

Author response: A detailed procedure for the DNA isolation is already present in the original manuscript. The content has been slightly rephrased to clarify and include all info.

In all the DNA extraction protocols, it would be good for the authors to mention how the purity and quantity of the extracted DNA was ascertain before performing the sequencing.

Author response: Procedure for DNA quantification prior to preparing libraries for genome sequencing has been included in Materials and Methods section 2.6. Following the described experimental procedures, we usually obtain good sequence data quality and DNA purity determinations are not routinely performed. 

Line 506: The authors mention samples of 2001, which does not appear in the materials and methods (See Table 1).

Author response: This is a typing error - sorry for this. It should be 1991 (not 2001) and has been corrected in the revised manuscript.

Reviewer 3 Report

In the manuscript “Antibiotic resistance and phylogeny of Pseudomonas spp.  isolated over three decades from chicken meat in the Norwegian food chain” Authors have determine the occurrence of antibiotic resistance in psychrotrophic Pseudomonas spp. collected through a time span of 26 years from retail chicken in Norway and characterize their genetic diversity, phylogenetic distribution and resistance gene reservoirs through whole genome sequence analyses. I found the subject of article interesting. Methodology used is in general correct and obtained results are promising. I recommend the manuscript for publication after addressing following issues:

  1. 16S rRNA gene sequenced and whole genome sequencing, why its was performed separately?
  2. Author should mention the program used for the phylogenetic tree.
  3. Reframe the sentence in line #219-221 in more clear way (Bacterial DNA (1 ul) isolated…………….…….recommendations was used as template).
  4. Correct the sentence in line 347-349 as: collection 4 (Year: 2017; n=38) contained the lowest and highest proportion of susceptible (5.3 %; n=2) and multi-resistant (28.9 %; n=11) isolates (Figure 2), respectively.
  5. Correct the sentence in line 350-351 as: the proportion of isolates susceptible to all antibiotics or resistant to ≥3 antibiotics were 0% and 53.8 %, respectively.
  6. Line 372, 533, correct the typo error ‘og’.
  7. Line 533, correct the typo error ‘om’.
  8. Line 576, remove word ‘of’.
  9. The typo and grammatical errors throughout the manuscript should be corrected. 
  10. References are not prepared according to author guidelines.

Author Response

Response to Reviewer 3 comments

In the manuscript “Antibiotic resistance and phylogeny of Pseudomonas spp.  isolated over three decades from chicken meat in the Norwegian food chain” Authors have determine the occurrence of antibiotic resistance in psychrotrophic Pseudomonas spp. collected through a time span of 26 years from retail chicken in Norway and characterize their genetic diversity, phylogenetic distribution and resistance gene reservoirs through whole genome sequence analyses. I found the subject of article interesting. Methodology used is in general correct and obtained results are promising. I recommend the manuscript for publication after addressing following issues:

Authors response: We thank the reviewer for a thorough review and for addressing valuable points to improve the manuscript. 

1. 16S rRNA gene sequenced and whole genome sequencing, why its was performed separately?

Authors response: The 16S rRNA gene sequencing was done to exclude non-Pseudomonas genus isolates from the collection and to obtain an overall identification and phylogenetic analyses as described in Materials and Methods section 2.2. and Results section 3.1. We had not the capacity or resources to perform whole genome sequencing on all isolates.  

2. Author should mention the program used for the phylogenetic tree.

Authors response: In section 2.3 it is stated that the CLC Main Workbench and the Neighbor Joining tree was used for construction of the phylogenetic tree based on 16s rRNA gene sequences. Core genome phylogenetic analysis was performed in Roary as stated in section 2.6.1. No changes were therefore made.

3. Reframe the sentence in line #219-221 in more clear way (Bacterial DNA (1 ul) isolated…………….…….recommendations was used as template).

Authors response: The sentence has been rephrased for clarification

4. Correct the sentence in line 347-349 as: collection 4 (Year: 2017; n=38) contained the lowest and highest proportion of susceptible (5.3 %; n=2) and multi-resistant (28.9 %; n=11) isolates (Figure 2), respectively.

Authors response: The sentence has been corrected as suggested by the Reviewer

5. Correct the sentence in line 350-351 as: the proportion of isolates susceptible to all antibiotics or resistant to ≥3 antibiotics were 0% and 53.8 %, respectively.

Authors response: The sentence has been corrected as suggested by the Reviewer

6. Line 372, 533, correct the typo error ‘og’.

Authors response: We were not able to find a typing error at the indicated lines.

7. Line 533, correct the typo error ‘om’.

Authors response: Corrected to "on"

8. Line 576, remove word ‘of’.

Authors response: Sorry, but we were not able to find a typing error at this line.

9. The typo and grammatical errors throughout the manuscript should be corrected. 

Authors response: We have gone through the manuscript using spell check and several typo and grammatical errors have been corrected.

10. References are not prepared according to author guidelines.

Authors response: References have been prepared according to the author guidelines in the revised manuscript

Round 2

Reviewer 1 Report

I have no further comments.

Reviewer 2 Report

My previous comments have been addressed, however, the authors should still read through the manuscript for minor language issues.